# Does Double Centrifugation Lead to Premature Platelet Aggregation and Decreased TGF-β1 Concentrations in Equine Platelet-Rich Plasma?

**DOI:** 10.3390/vetsci6030068

**Published:** 2019-08-21

**Authors:** Sarah R. T. Seidel, Cynthia P. Vendruscolo, Juliana J. Moreira, Joice Fülber, Tatiana F. Ottaiano, Maria L. V. Oliva, Yara M. Michelacci, Raquel Y. A. Baccarin

**Affiliations:** 1Departamento de Clínica Médica, Faculdade de Medicina Veterinária e Zootecnia, Universidade de São Paulo, São Paulo 05508-270, Brazil; 2Departamento de Bioquímica, Escola Paulista de Medicina, UNIFESP, São Paulo 04023-062, Brazil

**Keywords:** platelet-rich plasma, equine, platelet aggregation, growth factor

## Abstract

Blood-derived autologous products are frequently used in both human and equine medicine to treat musculoskeletal disorders. These products, especially the platelet-rich plasma (PRP), may contain high concentrations of growth factors (GFs), and thus improve healing in several tissues. Nevertheless, the procedures for preparation of PRP are currently non-standardized. Several protocols, which are based on distinct centrifugation patterns (rotation speed and time), result in PRPs with different characteristics, concerning platelet and GFs concentrations, as well as platelet activation. The aim of the present study was to compare two different protocols for PRP preparation: protocol (A) that is based on a single-centrifugation step; protocol (B), which included two sequential centrifugation steps (double-centrifugation). The results here reported show that the double-centrifugation protocol resulted in higher platelet concentration, while leukocytes were not concentrated by this procedure. Although platelet activation and aggregation were increased in this protocol in comparison to the single-centrifugation one, the TGF-β1 concentration was also higher. Pearson’s correlation coefficients gave a significant, positive correlation between the platelet counts and TGF-β1 concentration. In conclusion, although the double-centrifugation protocol caused premature platelet aggregation, it seems to be an effective method for preparation of PRP with high platelet and TGF-β1 concentrations.

## 1. Introduction

In the last decade, hemoderivative therapeutics has become a suitable alternative to treat musculoskeletal disorders in both human and veterinary medicine. Particularly, the platelet-rich plasma (PRP) has shown beneficial effects on several joint diseases, which help to relieve pain and improve the articular structures (reviews in [1,2,3,4]).

A systematic review on the efficacy of PRP in equine and human orthopedic therapeutics [1] concluded that, although many equine studies yielded positive results, the same is not true for human clinical trials. Furthermore, beneficial results for both species were more frequently observed in studies with a high risk of bias, which lead the authors to conclude that the use of PRP has still not shown strong positive effects in clinical scenarios.

Campbell et al. [2] have demonstrated, through meta-analysis and systematic review, that many studies, including comprehensive and meticulous ones, lack detailed information regarding the procedure for PRP preparations (centrifugation steps) and their characteristics (platelet counts, growth factor concentrations, and administered volumes).

On the other hand, PRP is an accessible and inexpensive source of growth factors [5], especially transforming growth factor β1 (TGF-β1). Therefore, it seems that the standardization of protocols to prepare PRP, as well as the careful characterization of PRP, is worthwhile.

Different authors have used distinct centrifugation patterns (rotation force, time, and number of centrifugation steps) to prepare PRP, possibly resulting in different final products [6]. Some protocols include two centrifugation steps, the second at higher force g. In the final step, the platelets are resuspended in a reduced plasma volume [7]. The platelet count in PRP can reach 13 times that of filtered apheresis platelet concentrates from whole blood, and the TGF-β1 concentration also increases, reaching an excess of 50 ng/mL [8]. However, Argüeles et al. [7] reported that the TGF-β1 concentrations were similar in PRPs that were obtained by either single- or double-centrifugation methods.

Although TGF-β1 is not the only growth factor present in PRP, it plays a significant role in musculoskeletal disorders, and it will be emphasized here. TGF-β is a multifunctional cytokine that belongs to the TGF superfamily, which includes three different mammalian isoforms (TGF-β 1 to 3) and bone morphogenetic protein (BMP). The isoform TGF-β1 was first identified in human platelets as a protein of 25 kDa, with a potential role in wound healing. Most of the immune cells secrete TGF-β1 [9].

In vitro and in vivo studies revealed that TGF-β ameliorates the deleterious effects of IL-1 on the cartilage matrix, counteracting both the IL-1 induced proteoglycan degradation and the suppression of synthesis [10]. Additionally, TGF-β regulates chondrocyte proliferation and differentiation [11] and induces the production of type II collagen and aggrecan [12]. Increased TGF-β1 concentrations (up to 10 or 100 ng/mL) correlate to increased proteoglycan and hyaluronan synthesis in cartilage [13] and meniscal [14] explants.

However, the role of TGF-β in joints remains controversial [15]. While TGF-β1 seems to favor chondrogenesis [16], multiple intra-articular injections of TGF-β1 may also promote osteoarthritis [17]. There are evidences that indicate that high doses of TGF-β may stimulate synovial proliferation and fibrosis, attracting inflammatory leukocytes to the synovial membrane, and thus inducing osteophyte formation [18,19].

These controversial results emphasize the need to standardize PRP preparation and to characterize the products, aiming at high platelet counts and high TGF-β1 concentrations. This is usually achieved by increasing centrifugal force, and/or time, but a limiting point is that platelet activation and aggregation also increases by increasing these centrifugal patterns. Platelet activation leads to TGF-β1 release, and the resulting preparation could be rich in platelets, but poor in TGF-β1. Thus, careful characterization of PRP is necessary, especially when intended for intra-articular administration, concerning the TGF-β1 concentration, to minimize the adverse effects [15].

The platelet-poor plasma (PPP) is the supernatant, which is usually discharged during PRP preparation. This fraction can be considered to be a byproduct and it has demonstrated positive effects when used in subcutaneous tissue during orthopedic surgery [20,21] and also as a single injection for chronic plantar fasciitis [22]. Nevertheless, PPP has been poorly explored in veterinary practice [23].

Therefore, the aim of the present study was to compare PRP that was prepared by two different protocols: a “single-centrifugation protocol”, based on one centrifugation step, and a “double-centrifugation protocol”, based on two centrifugation steps, in sequence. The PRP and PPP preparations were analyzed for platelet counts, platelet activation, and TGF-β1 concentration.

## 2. Materials and Methods

### 2.1. Animals

The present study was approved by the Faculdade de Medicina Veterinária e Zootecnia of the Universidade de São Paulo’s Ethics Committee on Animal Use (CEUA/USP; 1209080715), and it was carried out in accordance with EC Directive 86/609/EEC for animal experiments http://ec.europa.eu/environment/chemicals/lab_animals/legislation_en.htm. Twelve gelding male horses, aged between three and five years, which were clinically healthy and free of any drugs for 15 days prior to the study, were included in the present study.

The sample size calculation was determined to provide an 80% statistical power mainly for TGF-β1, and platelet concentration to detect a difference of 30% between the groups, with a two-sided alpha level of 0.025 and β = 0.20 based on two-way analysis of variance.

### 2.2. Blood Collection

Blood was collected from jugular vein following aseptic preparation of the site. For the single centrifugation protocol (see below), blood was collected while using a 21-gauge needle and a 10 mL syringe, and transferred to a Falcon tube (15 mL) containing 1 mL of 3.8% sodium citrate as anticoagulant. For the double centrifugation protocol, a 21-gauge needle and seven vacutainer tubes containing 3.8% sodium citrate were used (4.5 mL; final dilution blood:anticoagulant was 9:1). After harvesting and homogenization of the blood, the final volume was aliquoted and transferred to three Falcon tubes (15 mL), 10.5 mL each.

### 2.3. Platelet Rich Plasma Preparation

Two different protocols were used for PRP preparation: a single and a double centrifugation procedure, as schematized in Figure 1. The products were analyzed.

The single centrifugation protocol that was used here was adapted from Ottaiano et al. [24], in which premature platelet aggregation is avoided. In brief, blood was collected in one tube and centrifugated at 140× *g* for 12 min. (room temperature), and the supernatant was collected was named “PRP-0”.

For the double centrifugation protocol [4], three 15 mL tubes were used, containing 10.5 mL of blood each. After collection, the blood samples were incubated at room temperature for 25 min., and then were centrifugated at 300× *g* for 5 min., followed by another 25 min. “rest period”. It was empirically observed that less platelet activation occurred when these “rest period” were included. The supernatants were collected, and one of them was reserved and named “PPP-1”, while the other two were submitted to a second 700× *g*, 15 min. centrifugation, followed by another “rest period” of 45 min. The supernatant from the second tube was collected, reserved, and named “PPP-2”. The pellet and remaining volume were gently homogenized using a pipette and named “PRP-1”. The supernatant of the third tube, named “PPP-0”, was collected and cleared by centrifugation at 1000× *g* for 10 min., to assure minimal platelet counting in the supernatant. This was use as a sample with 100% light transmission in aggregometry testing, as described below.

### 2.4. Platelet and Leukocyte Counts

A complete blood count was performed on each blood sample. Leukocyte (WBC) counts were completed in the PRP-0, PRP-1, and PPP-1 sample while using a flow cytometry hematology system (ADVIA 2120i, Siemens – Erlangen, Germany).

Platelet counts were performed in whole blood (baseline values) samples and in the PRP-0, PRP-1, PPP-1, and PPP-2 samples while using a Neubauer chamber. Platelet concentrations were described as a ratio between platelet count for each sample and the platelet count for the whole blood sample.

### 2.5. Aggregometry Test

Platelet aggregation responses were evaluated while using an aggregometer (Chrono-Log Corporation, model 700) and monitored by the turbidimetric method developed by Born and Cross [25] using Agrolink software (Chrono-Log Corporation, Havertown, Pennsylvania, USA). Prior to this test, the PPP-0 was used as sample with 100% light transmission. To perform this test, 500 µL of PPP-0 was used as a blank. Likewise, 498 µL of each sample were placed in a 6-mm-wide siliconized cuvette with a stirring bar magnet and maintained at 37 °C. The samples were then identified with the software and baselines were adjusted to 0%. Two µL of agonist collagen type I (Chrono-Log Corporation) were added to the cuvettes, which were agitated. The aggregation curve was then observed for a 6-min. period.

This test was performed in all samples, except PPP-2, because, due to its low platelet counts, PPP-2 did not gave reliable results. The agonist alone was added as a control. All of the samples were stored at −80 °C until further analysis and TGF-β1 determination.

### 2.6. Determination of TGF-β1 Concentrations

TGF-β1 concentrations were quantified in all samples by ELISA, while using a commercially available human TGF-β1 kit (Quantikine ELISA, R&D Systems, Inc; Minneapolis, MN, USA, DB100B), previously validated for use in equine samples [26,27], in accordance with manufacturer instructions. All of the samples were run in duplicates.

After thawing and homogenization, TGF-β1 of each sample was activated: aliquots (20 µL) were placed in microcentrifuge tubes (200 μL), and 10 µL of 1 M HCl were added, which were then homogenized and incubated at room temperature. After 10 min., 10 µL of 1.2 M NaOH and 160 µL of RD5-53 diluent (R&D Systems, Inc; Minneapolis, MN, USA, part #895587) were added to each sample. RD5-53 diluent (50 µL) was added to all the wells of a 96-well anti-TGF-β1 antibody coated plate (TGF-β1 plate). Standards or samples (50 µL) were then added to the appropriate wells and then incubated for 2 h at room temperature with gentle agitation. The solution was discarded and the plate was washed four times with wash buffer solution (R&D Systems, Inc; Minneapolis, MN, USA, part #895126). Next, 100 µL of anti-TGF-β1 polyclonal antibody conjugated with HRP (R&D Systems, Inc; Minneapolis, MN, USA, part #893003) were added to each well, which were then covered by a film and then incubated again for 2 h at room temperature. The washing procedure was repeated, as previously described, and then 100 µL of substrate solution were added to each well. The plate was then covered with aluminum foil and incubated for 30 min. at room temperature. Finally, 100 µL of stop solution was added to each well. The wells were homogenized and absorbances were read while using a microplate reader at 450 nm.

### 2.7. Statistical Analyses

The data were analyzed for normality while using Kolmogorov–Smirnov tests. Afterwards, an analysis of variance (ANOVA), followed by Tukey-Kramer or Bonferroni tests, were used to compare parametric data. Non-parametric data were analyzed via the Friedman test. Correlation analyses were performed while using coefficients of determination and Pearson’s correlation coefficients. Statistical analyses were performed using GraphPad Instat 3 (GraphPad Software, San Diego, CA, USA). Statistical significance was set to *p* < 0.05 and confidence interval (CI) of 95%.

## 3. Results

### 3.1. Platelet and Leukocyte Counts and Platelet Concentration Times

The baseline platelet counts for the horses that were used in the present study was 180 × 10^3^/μL. Figure 2 shows that the platelet counts were significantly increased in PRP-0, obtained by the single-centrifugation protocol, and also in the products that were obtained by the first and the second centrifugation steps in the double-centrifugation protocol (PPP-1, and PRP-1, respectively) (*p* = 0.0001). For PRP-0, the mean platelet count was 494 ± 157 × 10^3^/μL (2.7 times the baseline), while for PPP-1 and PRP-1, the platelet counts were 756 ± 228 × 10^3^/μL (4.2 times) and 1371 ± 423 × 10^3^/μL (7.6 times), respectively. These data show that higher platelet counts were obtained in the products that were prepared by the double-centrifugation protocol (PPP-1 and PRP-1), as compared to the lower values that were obtained by the single centrifugation procedure (PRP-0). As expected, the platelet counts in PPP-2 (supernatant of the second centrifugation step) was lower than the baseline.

In contrast, the leukocyte counts decreased in all analyzed products (PRP-0, PRP-1, and PPP-1) relative to the baseline (Figure 2), which indicated that the platelet-rich plasma was not leukocyte-rich.

### 3.2. Platelet Aggregation

Aggregometry tests revealed significant differences among the groups (Figure 2). PRP-0 have shown high aggregation (110%, relative to baseline), with small variations among different samples. This indicates that this protocol did not induce previous platelet activation. In contrast, the aggregation response was significantly lower (*p* = 0.0001) in the samples that were prepared by the double-centrifugation protocol: PPP-1 = 90.9% and PRP-1 = 66.8%. This indicates that a certain degree of platelet activation/aggregation had already occurred during the experimental procedure.

### 3.3. TGF-β1 Concentrations

Figure 3A shows that the PRP-1 samples contained TGF-β1 concentrations that were significantly higher (12,407 pg/mL) than those that were found in PRP-0, PPP-1, and PPP-2 (*p* = 0.0001). The mean TGF-β1 concentrations in PRP-0 (5537 pg/mL) and PPP-1 (5539 pg/mL) were similar to each other, while PPP-2 presented the lowest concentrations (mean 2942 pg/mL) (Figure 3A).

Figure 3B shows that there is a positive correlation between platelet count and TGF-β1 concentrations (R^2^ = 0.74; Pearson’s correlation coefficient = 0.86).

Table 1 shows that the TGF-β1 concentration was proportional to the platelet counts in PRP-0, PPP-1, and PRP-1. The only exception was PPP-2 (the supernatant of double centrifugation protocol), which presented low platelet counts and relative higher TGF-β1 concentrations. 

Table 2 shows the TGF-β1 concentrations in the final volume of each product analyzed (across samples PRP-0, PRP-1, PPP-1, and PPP-2).

## 4. Discussion

The aim of the present study was to establish a protocol to prepare PRP with high platelet counts, as well as high TGF-β1 concentration. PRPs with such characteristics are supposed to be more useful for intra-articular administration in treating joint diseases.

Two different protocols were tested, and the products thus obtained were characterized, concerning platelet counts, TGF-β1 concentrations, and premature platelet activation/aggregation.

Our results have shown that the first protocol, based on a single centrifugation step, avoided premature platelet activation/aggregation, but resulted in a preparation with relatively low platelet and TGF-β1 concentrations (PRP-0).

In contrast, the final product that was obtained by the double-centrifugation protocol—PRP-1—presented high platelet counts and high TGF-β1 concentration, but some degree of premature platelet activation occurred. Leukocyte counts were not concentrated.

PPP-1, the product of the first centrifugation step in the double centrifugation protocol, was similar to PRP-0. This product could be considered to be a “PRP”, since it presents the minimum platelet concentration required to be considered a “PRP” by Fortier et al. [19] (4.2 times).

Table 3 summarizes the main characteristics of our preparations and of the preparations described in the literature.

The platelet concentrations in PPP-1 and PRP-1 were higher than many other preparations for equines [7,28,29,31], and was also higher than the concentration that was obtained by Hessel et al. [30] while using the E-PET ^®^ device.

The PRP-1, obtained after the second centrifugation step, resulted in a platelet concentration of 7.6 times the baseline. This concentration was higher than most of other studies, both in equines and human (Table 3), but it was slightly lower than those reported by Textor et al., using double centrifugation [32] that achieved weight times concentration, and Sutter et al. [8] that achieved 13 times and nine times concentrations, respectively, by apheresis/filtration and buffy coat methods. Nevertheless, apheresis only gave lower platelet concentrations (5 times) [8].

The presence of leukocytes in PRP may be useful or harmful, depending on the planned use and treatment objective [38]. In the present study, PRP was aimed for intra-articular injection. In this case, leukocytes are undesirable, because it is already known that patients who receive leukocyte-rich PRP present more joint swelling and pain [39], besides other deleterious effects [2,33]. Leukocyte were not concentrated in none of our products—PRP-0, PPP-1, and PRP-1. Nevertheless, it is important to note that the leukocytes count in PRP-1 were higher than those that were observed by Vendruscolo et al. [29] in their study of ten different protocols for double centrifugation. We also found higher leukocyte counts than those that were reported by Pereira et al. [35]. These results might be explained by differences in the plasma collection after centrifugation (although always avoiding the “buffy coat”).

The leukocyte concentration in our PRP-0 was also very low, in contrast to the results that were reported by Fontenot et al. [28], who also used a single-centrifugation protocol, but at higher speed, resulting in leukocytes-rich plasma. Of course, the growth factor composition of PRP is affected by the presence of other cell types [34]. 

In PRP preparation, besides the concentration of platelets, its functionality is also an issue, because platelets that were prematurely activated have possibly released their growth factors. This point was evaluated here by an aggregometry test, which was induced by collagen as exogenous agonist. By the single-centrifugation protocol, the platelets were not activated, in contrast to the double-centrifugation protocol, in which 90% of platelets responded after the first centrifugation step (PPP-1), and only 66.8% responded after the second centrifugation step (PRP-1). These results suggest that increased centrifugation speed and time could lead to platelet activation, which is in agreement with other authors [40]. Argüeles et al. [7] observed higher platelet activation after the first centrifugation step, which was possibly due to the presence of red blood cells in these samples, while Kingston et al. [41] have shown that sodium citrate partially inhibits platelet aggregation.

In the present study, collagen was used as exogenous agonist. Reversible responses are obtained [42], although other agonists may be also used, such as epinephrine and ADP, in contrast to collagen that induces irreversible aggregation and release of growth factors, mainly TGF-β1 and PDGF-BB in a dose-dependent fashion, up to 10% of the estimated total platelet content [32].

The TGF-β1 concentrations were similar in PRP-0 (single-centrifugation) and PPP-1 (first centrifugation step in the double-centrifugation protocol), despite their differences in platelet counts. This could be due to the premature platelet activation and partial TGF-β1 release in the preparation of PPP-1.

The highest TGF-β1 concentrations were found in PRP-1 (double-centrifugation procedure), again demonstrating a correlation between platelet counts and TGF-β1 concentration. This result agrees with other studies on PRP in equines [8,33,34,36]. PPP-2, which is the supernatant of the double-centrifugation protocol, contained low platelet counts and high TGF-β1 concentrations, possibly due to the premature platelet activation and TGF-β1 release during the second centrifugation step.

Table 3 also shows that other authors [32,37] obtained preparations with high platelet counts, but low TGF-β1 concentrations in comparison to ours. According to Textor and Tablin [37], treatment with bovine thrombin as agonist or with 10% calcium chloride increased TGF-β1 concentrations, which indicated that TGF-β1 was released by platelet activation. Argüelles et al. [7] obtained a preparation with low platelet counts, but high TGF-β1 concentrations (after 10% calcium chloride activation), and the authors concluded that single centrifugation was “more efficient”, meaning that less premature platelet activation occurred. This result is similar to ours. 

Pereira et al. [35] compared seven different double-centrifugation protocols, which concentrated the platelets from 4.1 to 5.4 times, and the TGF-β1 concentrations achieved our PRP-1 values in only one of the protocols (Table 3). The values that were obtained in all of these studies, as well as in the present report, were lower than those that were reported by Sutter et al. [8] by apheresis, followed by filtration.

As platelet activation is necessary to release growth factors, the avoidance of premature activation guarantees the presence of growth factors at the time of PRP use. Most of the growth factors have short half-lives after release (minutes to a few hours). If activated platelets are not used in an adequate timeframe, the growth factors become inactivated before application to the target tissue [43]. Consequently, premature platelet activation or a delay in use can result in poor therapeutic efficacy.

It should be emphasized that the procedures described here permit the preparation of PRPs with different characteristics, which could be useful in treating different clinical joint conditions. For example, PRP-0 may be useful to treat large joints, in which the injection volume is not an issue, while PRP-1, whereas more laborious and time-consuming, may be appropriate to treat small joints.

## 5. Conclusions

The present study has compared two procedures for preparation of PRP, one that is based on a single-centrifugation step and one based on double-centrifugation. In the single-centrifugation procedure, no premature platelet activation occurred, but platelets and TGF-β1 were less concentrated. The double-centrifugation protocol resulted in high platelet and TGF-β1 concentrations, but a certain degree of activation occurred, which was evidenced by both aggregometry tests and the presence of TGF-β1 in supernatant. All of our preparations were platelet-rich but not leukocyte-rich. Taken together, these results emphasize the need to carefully characterize the PRP obtained by any procedure, to produce reliable and reproducible results.

## Figures and Tables

**Figure 1 vetsci-06-00068-f001:**
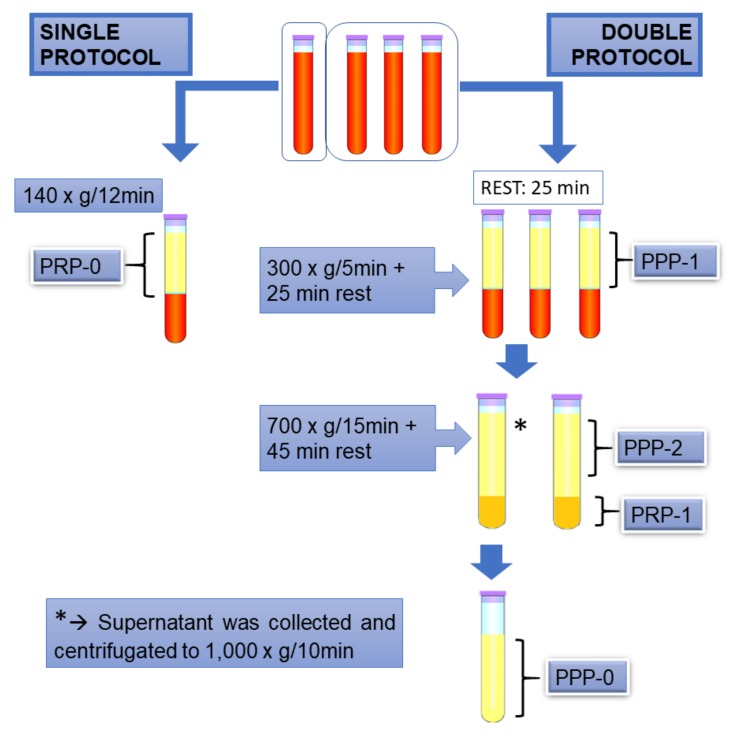
Illustrative scheme of the protocols used in the present study. Platelet-rich plasma (PRP)-0, PRP obtained by the single protocol; platelet-poor plasma (PPP)-1, platelet poor plasma obtained in the first centrifugation step of double protocol; PPP-2, supernatant of the second centrifugation step of double protocol; PRP-1: PRP obtained by the double protocol; PPP-0, supernatant after centrifugation of PPP-2.

**Figure 2 vetsci-06-00068-f002:**
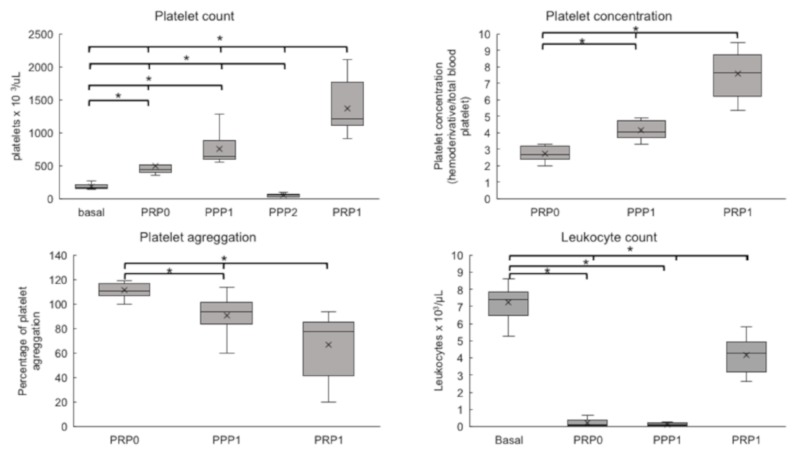
Platelet counts, Platelet concentrations, Platelet aggregation (%), and Leukocyte counts of PRP and PPP preparations. The figure shows results as Box plots, representing medians (horizontal central lines), means (x), 25th and 75th percentiles (rectangles, first and third quartiles), and minimal and maximal values (vertical lines) for each experimental group. Basal, whole blood; PRP0, PRP obtained by the single protocol; PPP1, platelet poor plasma obtained in the first centrifugation step of double protocol; PPP2, supernatant of the second centrifugation step of double protocol; PRP1, PRP obtained by the double protocol; PPP0, supernatant obtained after centrifugation of PPP2. * Differences statistically significant (*p* < 0.05).

**Figure 3 vetsci-06-00068-f003:**
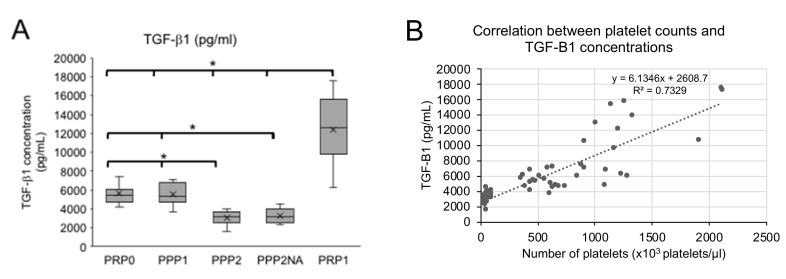
Transforming growth factor β1 (TGF-β1) concentrations (**A**) and correlation between Platelet counts and TGF-β1 concentrations (**B**). The results in (**A**) are shown as Box plots, representing medians (horizontal central lines), means (x), 25th and 75th percentiles (rectangles, first and third quartiles), and minimal and maximal values (vertical lines) for each experimental group. PRP0, PRP obtained by the single protocol; PPP1, platelet poor plasma obtained in the first centrifugation step of double protocol; PPP2, supernatant of the second centrifugation step of double protocol; PPP2NA, PPP2 with no agonist; PRP1, PRP obtained by the double protocol. * Differences statistically significant (*p* < 0.05).

**Table 1 vetsci-06-00068-t001:** Ratio between TGF-β1 concentrations and platelet counts obtained from platelet-rich plasma 0 (PRP-0) and 1 (PRP-1), and platelet-poor plasma 1 (PPP-1) and 2 (PPP-2).

Animal	PRP-0	PPP-1	PPP-2	PRP-1
1	10.9	7.2	27.6	11.6
2	6.4	7.9	55.8	8.2
3	11.5	11.3	61.0	13.5
4	11.9	6.1	32.5	5.1
5	12.0	12.2	92.9	12.6
6	8.5	4.7	39.1	8.3
7	15.5	4.4	34.6	5.6
8	16.2	7.2	57.8	13.0
9	11.2	7.2	88.2	10.5
10	16.4	6.8	98.8	10.1
11	12.3	10.0	74.5	6.3
12	9.7	8.4	76.0	8.2
Mean	11.9	7.8	61.6 *	9.4
SD	3.0	2.4	24.7	2.9

PRP-0, PRP obtained by the single protocol; PPP-1, platelet poor plasma obtained in the first centrifugation step of double protocol; PPP-2, supernatant of the second centrifugation step of double protocol; PRP-1: PRP obtained by the double protocol; PPP-0, supernatant after centrifugation of PPP-2. * Difference statistically significant (*p* < 0.05).

**Table 2 vetsci-06-00068-t002:** TGF-β1 concentrations (pg) in total volume (mL) obtained in platelet-rich plasma 0 (PRP-0), 1 (PRP-1), and platelet-poor plasma 1 (PPP-1) and 2 (PPP-2).

Product	Total Volume (mL)	Mean TGF-β1 (pg/mL)	Mean TGF-β1 Total
PRP-0	6.5	5537	35,992
PPP-1	6.5	5539	36,006
PPP-2	4.5	3102	13,960
PRP-1	2.0	12,407	24,815

PRP: Platelet-rich plasma; PPP: Platelet-poor plasma; TGF-β1: Transforming growth factor beta 1.

**Table 3 vetsci-06-00068-t003:** PRP characterization.

Authors	Specie	Method	Agonist	Platelet Count × 10^3^/μL	Leukocyte Count × 10^3^/μL	TGF-β1 pg/mL
Arguelles et al. [7]	Equine	Single	Calcium chloride	229	4.1	9400
		Single	Calcium chloride	228	3.1	10,300
		Double	Calcium chloride	272	8.4	10,500
		Double	Calcium chloride	191	0.93	9900
Sutter et al. [8]	Equine	Apheresis + filtration	-	2172	61.2	57,900
		Buffy coat	-	1472	32.5	15,300
		Apheresis	-	855	33.7	23,600
Fontenot et al. [28]	Equine	Single	-	267	-	-
		Single	-	399	-	-
		Single	-	433	-	-
		GenesisSC^®^	-	359.2	-	-
Vendruscolo et al. [29]	Equine	Double	-	343.9	4.8	453.10
		Double	-	363.6	2.4	506.23
		Double	-	319.1	1.2	839.23
		Double	-	344.9	4.1	541.15
Hessel et al. [30]	Equine	Angel^®^	-	320.3	9.1	660
		ACP^®^	-	183.2	0.6	850
		E-PET^®^	-	533.3	11.0	1700
		GPS^®^	-	761	40.6	680
		Double	-	310.4	18.2	1580
Carmona et al. [31]	Equine	Double	Calcium chloride	250	8.68	12,515
Textor et al. [32]	Equine	Double	Collagen	1765	-	2219
		SmartPReP 2^®^	Collagen	951	-	3707
McCarrel et al. [33]	Equine	SmartPReP 2^®^	-	513	6.9	2000
Sundman et al. [34]	Human	Arthrex ACP^®^	-	361	0.6	20,000
		Biomet GPS III^®^	-	701	23.7	87,000
Pereira et al. [35]	Equine	Double	-	4.1× *	-	14,053
		Double	-	4.7× *	-	7634
		Double	-	4.6× *	-	7198
		Double	-	4.8× *	-	8796
		Double	-	4.8× *	-	10,004
		Double	-	5.4× *	-	10,518
		Double	-	4.5× *	-	12,397
McLellan et al. [36]	Equine	Double	Calcium gluconate	-	-	22,640
Textor & Tablin [37]	Equine	Double	Autologous thrombin	770	-	3263
		Double	Bovine thrombin	770	-	9528
		Double	Calcium chloride	770	-	8808
		Double	Freeze-thaw	770	-	10,928
Our Results—PRP-0	Equine	Single	Collagen	494	0.23	5537
PPP-1	Equine	Double—1st step	Collagen	756	0.13	5539
PRP-1	Equine	Double—2nd step	Collagen	1371	4.18	12,408

TGF-β1: Transforming growth factor beta 1. *: Platelet concentration (times).

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
