# Peer review of "Does Double Centrifugation Lead to Premature Platelet Aggregation and Decreased TGF-β1 Concentrations in Equine Platelet-Rich Plasma?"

_vetsci, 2019, doi:10.3390/vetsci6030068_

Round 1
Reviewer 1 Report
Overall the premise for the study, study design and methods are good. There is some scope for improvement to the readability of the paper and presentation of the data. Some of the language is a little conversational and casual. Sentence structure is occasionally clumsy. Overall style would be improved by review by someone with attention to English grammatical style. Avoid doubling up when referencing previous studies by mentioning the author name and the number reference. More detailed comments are listed below.
Abstract
A succinct and readily understandable summary of the paper.
Introduction
The introduction could be shortened. Although it is important to appreciated the potential benefits of different characteristics of PRP products this could be presented in a more succinct fashion. Move the focus away from the controversial role of PRP, and towards justifying the reason for the study.
Methods
Comprehensive and protocol is understandable.
Statistics appropriate for dataset. It would be good practice to include confidence intervals as well as p values. How did you determine the number of animals required for this study (power calculation)? Or if it was a pilot study mention this.
Results
Written text generally easy to follow. Please add p values and confidence intervals to relevant parts of the text/ legends.
Consider requirement for decimal point accuracy in larger numbers e.g. line 194, table 2
Figure 1: Re-define acronyms in legend. I found it difficult trying to remember the difference between PPP, PRP without going back to the text. The figure should be standalone so it can be understood without the supporting text. Consider expanding the legend.
Figure 2: Use a standardised symbol for statistical significance, it is not clear what the comparison has been. Mention in figure legend. For box plots provide more detail about which is mean, median etc.
Figure 3: Increase size of figures. Again use same sign for statistical significance and document the comparison that has been made within the figure legend inc P values and confidence intervals. This needs changing throughout manuscript (in tables too). Complete definition of acronyms (PPPNA).
Table 2: Is total TGFB1 important? What are the limitations e.g. volume for joint injection?
Discussion
The discussion is long and readability could be improved. I found it difficult to just pick out the important findings which were very well communicated in the abstract.
How much in comparing to previous studies will the results be impacted by the horse population chosen? Is there likely to be much variation between individuals, breeds etc in the measured outcomes?
I felt the structure of the discussion drew the focus away from your study and onto other studies. It is important to add context of previous literature but be careful not to lose the readers ability to follow your train of thought. Particularly the first sentence of each paragraph is important to communicate the theme of the paragraph. For example you could consider turning the structure of the paragraphs around to begin with your finding and then summarise how this compares with previous results.
Avoid repetition of specific results values in the discussion (put all the key data points into one table and reference the table in the discussion). Try and provide more of an overview of the important findings. Make links to the clinical implications of the findings- what recommendations you would make based on these results for use in clinical practice?
Author Response
Reviewer 1 -
Overall the premise for the study, study design and methods are good. There is some scope for improvement to the readability of the paper and presentation of the data. Some of the language is a little conversational and casual. Sentence structure is occasionally clumsy. Overall style would be improved by review by someone with attention to English grammatical style. Avoid doubling up when referencing previous studies by mentioning the author name and the number reference. More detailed comments are listed below.
- The language and grammatical style were revised and modified accordingly
Abstract
A succinct and readily understandable summary of the paper.
Introduction
The introduction could be shortened. Although it is important to appreciated the potential benefits of different characteristics of PRP products this could be presented in a more succinct fashion. Move the focus away from the controversial role of PRP, and towards justifying the reason for the study.
- PRP characteristics were described in a more succinct fashion. A new paragraph about PPP was included, attending to the Reviewer’s 2 suggestion.
Methods
Comprehensive and protocol is understandable.
Statistics appropriate for dataset. It would be good practice to include confidence intervals as well as p values. How did you determine the number of animals required for this study (power calculation)? Or if it was a pilot study mention this.
- Confidence intervals and p values included (line 172). A paragraph was included to explain how the number of horses was determined (lines 94-96).
Results
Written text generally easy to follow. Please add p values and confidence intervals to relevant parts of the text/ legends.
- p values and confidence intervals included in legends.
Consider requirement for decimal point accuracy in larger numbers e.g. line 194, table 2
- Decimal numbers removed from text and from Table 2.
Figure 1: Re-define acronyms in legend. I found it difficult trying to remember the difference between PPP, PRP without going back to the text. The figure should be standalone so it can be understood without the supporting text. Consider expanding the legend.
- Acronyms defined. Legend expanded.
Figure 2: Use a standardised symbol for statistical significance, it is not clear what the comparison has been. Mention in figure legend. For box plots provide more detail about which is mean, median etc.
- Symbols were modified to indicate differences statistically significant, and mentioned in the figure legend. Details on box plot are given in figure legends.
Figure 3: Increase size of figures. Again use same sign for statistical significance and document the comparison that has been made within the figure legend inc P values and confidence intervals. This needs changing throughout manuscript (in tables too). Complete definition of acronyms (PPPNA).
- The size of the figures was increased, and the symbols were modified. PPPNA was defined.
Table 2: Is total TGFB1 important? What are the limitations e.g. volume for joint injection?
- It has already been shown that TGF-B1 is important to treat musculoskeletal diseases, although the ideal doses are not already known. The volume limitations for injection depend on the joint size. For horses, it may vary between 1 ml (for instance, distal intertarsal joint), and 20 mL (femorotibial joint). Thus, to achieve the needed final concentration, high TGF-B1 in PRP preparations are desirable.
Discussion
The discussion is long and readability could be improved. I found it difficult to just pick out the important findings which were very well communicated in the abstract.
- The Discussion was revised and shortened. We hope the Discussion will now meet your requirements.
How much in comparing to previous studies will the results be impacted by the horse population chosen? Is there likely to be much variation between individuals, breeds etc in the measured outcomes?
- A Table (Table 3) was included in Discussion comparing our results to other reported in the literature. Our results were compared with others (pages 8-10).
The aim of the present study was to investigate the effects of PRP preparation procedures – single vs. double centrifugation. We did not aim to investigate the effects of animal ages, breeds, and other biological differences. This point is stressed in Abstract, Introduction and Discussion.
I felt the structure of the discussion drew the focus away from your study and onto other studies. It is important to add context of previous literature but be careful not to lose the readers ability to follow your train of thought. Particularly the first sentence of each paragraph is important to communicate the theme of the paragraph. For example you could consider turning the structure of the paragraphs around to begin with your finding and then summarise how this compares with previous results.
- The Discussion was thoroughly modified. We hope the present form will meet your requirements.
Avoid repetition of specific results values in the discussion (put all the key data points into one table and reference the table in the discussion). Try and provide more of an overview of the important findings. Make links to the clinical implications of the findings- what recommendations you would make based on these results for use in clinical practice?
- Repetition was removed. Comments on the use to treat joint diseases were included (lines 317-320).

Reviewer 2 Report
The manuscript by Siedel et al., is a well written description about a standardized method to prepare platelet rich plasma (PRP) for use in horse. This is well described and presented, however, some minor revision is suggested, including language.
Line 42. PRP preparation and their characteristics…
Line 53. Keep only TGF-B1
Lines 56-60. Indicate references
Lines 67-71. Detail if is related to all TGF-B or only TGF-B1
Line 80. Please consider change ..So.. for Therefore
Line 82. There is the first mention to PPP, but there is not description about it and its importance.
Introduction: Consider describe PPP, as an alternative product of sub-product and why is mentioned as relevant.
Line 90. How the number horses were estimated for this study? Did you find differences between ages?
Figure 1. Since single protocol was based on Ottatiano et al., could be valuable know if the double protocol is original.
1000 should be 1,000
G is capital or minuscule?
Review the distances between tubes, especially in the non-centrifugated tubes.
Consider deleted “COLLECT”
Explain the asterisk
Line 107. Check if 140G or 141.
109. Explain why the resting time.
Lines 112-118. Revise redaction.
Line 144. Use microcentrifuge tube
Figure 3. What is PPP2NA?
Figures 2 and 3. Please consider increase the size of letters in axes.
Lines 203-206. Revise redaction.
Discussion
Lines 217 – 222. Consider English revision
Lines 231- 235. Is PPP1 a PRP? Is the answer is right why is called PPP? Explain if this is an alternate product of PRP1 and if could be also used in treatment.
Discussion. Please consider include a table that summarized your results and those obtained in other studies.
Author Response
Reviewer 2 -
The manuscript by Siedel et al., is a well written description about a standardized method to prepare platelet rich plasma (PRP) for use in horse. This is well described and presented, however, some minor revision is suggested, including language.
Line 42. PRP preparation and their characteristics…
- Modified, as suggested.
Line 53. Keep only TGF-B1
- Modified, as suggested.
Lines 56-60. Indicate references
- Indicated, as suggested.
Lines 67-71. Detail if is related to all TGF-B or only TGF-B1
- Details were given.
Line 80. Please consider change ..So.. for Therefore
- Modified, as suggested
Line 82. There is the first mention to PPP, but there is not description about it and its importance.
- A paragraph was included (lines 77-80), with new references (20-23).
Introduction: Consider describe PPP, as an alternative product of sub-product and why is mentioned as relevant.
- see above.
Line 90. How the number horses were estimated for this study? Did you find differences between ages?
- A paragraph was included to explain how the number of horses was determined (lines 94-96). The age range was kept small – all animals were 3-5 years old – to avoid age differences.
Figure 1. Since single protocol was based on Ottaiano et al., could be valuable know if the double protocol is original.
1000 should be 1,000
G is capital or minuscule?
Review the distances between tubes, especially in the non-centrifugated tubes.
Consider deleted “COLLECT”
Explain the asterisk
- The double protocol was developed by our group, and previously published in 2014 (reference 4). This reference was included in the text (line 115). 1,000 corrected; g minuscule; layout revised; “collect”deleted; asterisk explained.
Line 107. Check if 140G or 141.
- (now line 114) 140 x g (Figure 1 was corrected).
109. Explain why the resting time.
- Explained – see lines 117-118.
Lines 112-118. Revise redaction.
- Revised, as suggested.
Line 144. Use microcentrifuge tube
- Corrected, as suggested.
Figure 3. What is PPP2NA?
- Defined.
Figures 2 and 3. Please consider increase the size of letters in axes.
- Increased, as suggested.
Lines 203-206. Revise redaction.
- Revised and modified (see lines 193-217).
Discussion
Lines 217 – 222. Consider English revision
- English revised
Lines 231- 235. Is PPP1 a PRP? Is the answer is right why is called PPP? Explain if this is an alternate product of PRP1 and if could be also used in treatment.
- In fact, PPP-1 could be considered a PRP, since its platelet counts were higher than “Basal”, and many authors consider “PRP” only when platelet counts increased at least 4 times relative to “Basal”. Nevertheless, other authors using double centrifugation protocols (although not exactly as ours) name this fraction PPP. We named our fractions accordingly. This point was discussed (lines 246-248; 292-295).
Discussion. Please consider include a table that summarized your results and those obtained in other studies.
- Table 3 was included (page 8, lines 253-254).
